# Behavioral Voluntary and Social Bioassays Enabling Identification of Complex and Sex-Dependent Pain-(-Related) Phenotypes in Rats with Bone Cancer

**DOI:** 10.3390/cancers15051565

**Published:** 2023-03-02

**Authors:** Daniel Segelcke, Jan Linnemann, Bruno Pradier, Daniel Kronenberg, Richard Stange, S. Helene Richter, Dennis Görlich, Nicola Baldini, Gemma Di Pompo, Waldiceu A. Verri, Sofia Avnet, Esther M. Pogatzki-Zahn

**Affiliations:** 1Department of Anesthesiology Intensive Care and Pain Medicine, University Hospital Münster, 48149 Münster, Germany; 2Department of Clinical Radiology, Translational Research Imaging Center, University Hospital Münster, 48149 Münster, Germany; 3Institute for Musculoskeletal Medicine, University Hospital Münster, 48149 Münster, Germany; 4Department of Behavioural Biology, University of Münster, 48149 Münster, Germany; 5Institute of Biostatistics and Clinical Research, University of Münster, 48149 Münster, Germany; 6Department of Biomedical and Neuromotor Sciences, University of Bologna, 40126 Bologna, Italy; 7Biomedical Science and Technologies and Nanobiotechnologies Laboratory, IRCCS Istituto Ortopedico Rizzoli, 40136 Bologna, Italy; 8Centro de Ciências Biológicas, Departamento de Ciências Patológicas, Universidade Estadual de Londrina, Londrina 86057-970, Brazil

**Keywords:** bone cancer, pain, home cage, rodent-specific behavior

## Abstract

**Simple Summary:**

Bone metastases are one of the most common complications in patients with advanced cancer that result in pain, which is usually severe, thereby significantly reducing the patient’s quality of life. Although preclinical pain research in rodents is improving, the pain phenotyping methods currently used have been criticized. This study aimed to identify in detail pain phenotypes of cancer-induced bone pain (CIBP) in both sexes of rats. CIBP in the splint bone on one side results in a distinct CIBP-related phenotype characterized by mechanical hypersensitivity, resting pain, and antalgic gait in both sexes. Progression of tumor growth leads to the establishment of the CIBP phenotype that appears earlier in male than in female rats and affects rat-specific social behaviors in both sexes. We demonstrate social transfer of pain in a bone cancer model in both sexes, resulting in mechanical and, in females, also heat hypervigilance in non-tumor bearing control rats.

**Abstract:**

Cancer-induced bone pain (CIBP) is a common and devastating symptom with limited treatment options in patients, significantly affecting their quality of life. The use of rodent models is the most common approach to uncovering the mechanisms underlying CIBP; however, the translation of results to the clinic may be hindered because the assessment of pain-related behavior is often based exclusively on reflexive-based methods, which are only partially indicative of relevant pain in patients. To improve the accuracy and strength of the preclinical, experimental model of CIBP in rodents, we used a battery of multimodal behavioral tests that were also aimed at identifying rodent-specific behavioral components by using a home-cage monitoring assay (HCM). Rats of all sexes received an injection with either heat-deactivated (sham-group) or potent mammary gland carcinoma Walker 256 cells into the tibia. By integrating multimodal datasets, we assessed pain-related behavioral trajectories of the CIBP-phenotype, including evoked and non-evoked based assays and HCM. Using principal component analysis (PCA), we discovered sex-specific differences in establishing the CIBP-phenotype, which occurred earlier (and differently) in males. Additionally, HCM phenotyping revealed the occurrence of sensory-affective states manifested by mechanical hypersensitivity in sham when housed with a tumor-bearing cagemate (CIBP) of the same sex. This multimodal battery allows for an in-depth characterization of the CIBP-phenotype under social aspects in rats. The detailed, sex-specific, and rat-specific social phenotyping of CIBP enabled by PCA provides the basis for mechanism-driven studies to ensure robustness and generalizability of results and provide information for targeted drug development in the future.

## 1. Introduction

Bone metastases are one of the most common complications in patients with advanced cancer, and pain is the main and most devastating symptom [1]. Cancer-induced bone pain (CIBP) is usually high, can occur at rest, increases under certain conditions and is associated with a significant impairment of patients‘ quality of life [2]. Interestingly, common analgesics are only partly able to reduce pain and pain-related symptoms and side effects are common; radiation therapy might help but is also limited in efficacy. More specific analgesics targeting the specific mechanisms of CIBP are currently missing. Consequently, the mechanisms underlying CIBP are increasingly studied in preclinical animal pain models [3].

Based on pathophysiological findings, CIBP appears to have unique features and must be distinguished from other cancer-induced [1] and chronic pain states [4,5], which are embedded in disease and treatment classification [1,6]. However, how these cancer-related features and mechanisms relate to pain and pain-related symptoms relevant for the QoL in patients is currently unclear.

Although preclinical pain research is improving, widely used animal models and the currently applied phenotyping approaches have been criticized [3,7]. Behavioral phenotyping of different pain entities, including CIBP, is based mainly on evoked, reflexive withdrawal methods [3]. Although well-established and standardized, they have endpoints based only on sensory perception and may reduce the complex clinical reality to one feature, limiting their clinical value. [8,9]. Additionally, factors that have been shown to influence the outcome of these behavioral tests, such as testing rodents in light periods, the presence of an experimenter, or the experimenter’s gender, may limit the internal and external validity of findings [10,11]. Recent clinical interventions use multidimensional therapeutic approaches [9] and aim to restore daily life activities without decreasing physiological sensory perception to external stimuli. Furthermore, pain states always have a social dimension [12], not reflected by evoked pain assessments. This mismatch between preclinical research and clinical application critically contributes to a lack of efficient translational pain research and therefore needs to be bridged [7].

Novel behavioral assays that detect clinically relevant symptoms of different pain entities are vastly underrepresented, and only a few studies show a multimodal approach to examining pain-associated behaviors [7,13,14]. This problem is also present in CIBP research [3]. In the past two decades, novel, video-based assays have been developed to address (1) functional aspects, such as pain-related changes in gait patterns [14,15], (2) non-evoked pain- (NEP) related behavior [16,17,18], or (3) changes in rodent-specific complex behavior (e.g., resting, ambulatory, social, and pain-related behaviors) [19]. In this context, monitoring behavior in home-cage settings seems promising to broaden behavioral studies and investigate the animals within their familiar environments. In this way, the physical contact between the animals and human experimenters is reduced, thereby minimizing any direct experimenter effects [10,20,21]. At the same time, it might become possible to detect more subtle behavioral changes that indicate pain-related behavior in more detail and/or with more reliability [11,19,22,23].

To bridge the translational gap in CIBP, we combined traditionally used, reflex-based withdrawal assays with mechanical and heat stimuli with video-based approaches for non-evoked pain-related behavior in both sexes. Additionally, we analyzed the complex rat-specific behavior of chronic CIBP in a video-based home-cage setting to identify pain phenotypes for the first time in unprecedented detail. We used a highly efficient and well-described CIBP model in rats by using Walker 256 mammary carcinoma cells inoculated into the tibia. We hypothesized that in addition to the known changes in reflex-based pain-related behavior in both sexes, an antalgic gait, non-evoked pain, and changes in complex rat-specific behavior develop time-dependently due to the induction of a bone tumor and its progression. By integrating multimodal datasets, including home-cage observation, we detected previously unknown sex differences in the CIBP phenotype and behavioral changes in cancer and sham animals relative to the bystander. This will help to improve future mechanism-driven research that can ensure imprinted robustness and generalizability of results and ultimately inform target-specific drug development.

## 2. Materials and Methods

The experiments in this study were reviewed and approved by the Animal Ethics Committee of the State Agency for Nature, Environment, and Consumer Protection North Rhine Westphalia (LANUV, Recklinghausen, Germany; 84-02.04.2017.A055). We reported on the study details according to the ARRIVE guidelines 2.0 [24] and were in accordance with the ethical guidelines for investigating experimental pain in conscious animals [25].

### 2.1. General

Male and female Sprague-Dawley (SD) rats (total *n* = 114 rats, age: 6–7 weeks, weight males: 280.1 ± 7.8 g (Mean ± SD), weight females: 192.3 ± 5.3) were kept in a 12/12 h day/night cycle with ad libitum access to food and water under conventional conditions (FELASA guidelines) (Figure 1A, General). According to their experimental group, the rats were housed in pairs. The experimental group allocation was random, and a blinded analysis of video-based behavior assessments was performed. Blinding to the withdrawal/reflex-based behavioral assays and pain model was not possible due to the testing conditions and visible signs, such as a visible tumor or a guarding behavior of the cancer-bearing limb of CIBP. Finally, the animals were euthanized by decapitation under deep isoflurane anesthesia at the end of the observation period of 18 days.

### 2.2. Culture of Walker 256 Cells

Walker 256 mammary gland carcinoma cells (Cell Resource Centre for Medical Research at Tohoku University (CRCTU), Sendai, Japan) were cultured in a 20 mL medium containing RPMI 1640 (Sigma-Aldrich, Steinheim am Albuch, Germany), antibiotic antimycotic solution (100×) (Sigma-Aldrich, St. Louis, MI, USA), glutamine (Merck, Rahway, NJ, USA), and fetal bovine serum (FBS, Capricon Scientific, Ebsdorfergrund, Germany). For the preparation of the medium, a standard bottle (500 mL) RPMI and 5 mL antibiotic antimycotic solution (100×), 5 mL glutamine solution, and 50 mL FBS solution were mixed and dispensed to the culture bottles.

When 80% confluence was achieved, the cells were transferred to a new culture flask. This was done twice a week under the above conditions. To generate a sham group, those cells dedicated for inoculation in sham animals were separated and heat deactivated (10 min/95 °C). To ensure that heat deactivation was successful, a sample of heat-deactivated cells was re-cultured and checked for (the absence of) cell proliferation over two days after deactivation. Heat deactivation functioned in every case, eliminating the need to exclude any sham animals due to methodological deficiencies. Cells of passage 15–20, starting from the supplied cells from the cell bank, were used for inoculation.

### 2.3. Pain Model: Cancer-Induced Bone Pain (CIBP)

Any surgery was performed on the right hindlimb of rats. Rats were initially anesthetized with 5% isoflurane in 100% oxygen; anesthesia was maintained with 1.5–2.0% isoflurane delivered through a nose cone during the whole procedure. To minimize the wound pain caused by skin incision, Metamizole (Vetalgin^®^ 500 mg/mL, MSD Tiergesundheit, Friesoythe, Germany) was administered subcutaneously (100 mg/kg BW) 30 min in advance of the surgical procedure.

After anesthesia and analgesia were established, the surgical area was shaved and disinfected by the use of Betadine^®^ (Aviro Health, Cape Town, South Africa). For inoculation with Walker 256 cells [26,27], a skin incision of 1 cm above the right knee joint was made with a scalpel (No. 11, 0.5 cm), and the proximal tibial bone was displayed. Using a 23 G needle, a hole was drilled into the right tibial epiphysis to access the intramedullary space. Afterward, the needle was removed and replaced by a Hamilton syringe that contained the cell suspension (4 × 10^5^ Walker 256 cells in 10 µL Hank’s solution). Next, the cell suspension was injected into the intramedullary space of the right proximal tibia. The syringe remained inside the bone for two minutes after each cell injection to ensure that the cell suspension would not leak out of the intramedullary space right after inoculation [26]. After removing the syringe, the hole was closed with bone wax (SMI, St. Vith, Belgium). The skin incision was sutured with a mattress suture of 7-0 Prolene^®^ (Ethicon, Raritan, NJ, USA). Finally, the skin wound was disinfected with Betadine^®^ again.

Sham (inoculation of heat-deactivated cancer cells) and Naive rats (received Hank’s solution only) were used to control the cancer cell inoculation (Figure 1A, CIBP).

### 2.4. Multidimensional Assessment of Pain-Related Behaviors

Reflex-based withdrawal behaviors for hypersensitivity assessment of the ipsilateral hind paw

#### 2.4.1. Paw Withdrawal to Von Frey Filaments (PWT)

The ascending stimulus method determined the punctate paw withdrawal threshold (PWT) [22] by application of calibrated Semmes-Weinstein von Frey filaments (Bioseb, Vitrolles, France; 14, 20, 39, 59, 78, 98, 147, 255, 588 mN bending force) to the plantar side of the right hind paw (Figure 1B). Rats were placed on a mesh grid, and covered by a transparent plastic box (dimensions 15 × 20 × 10 cm, H × W × H). After a habituation period of 15 min, the filaments were applied in ascending order until the occurrence of withdrawal responses or reaching of the cutoff limit of 588 mN. If so, 588 mN was regarded as PWT. The median force of three trials leading to a response was considered as the PWT to mechanical stimuli.

#### 2.4.2. Paw Withdrawal Latency to Heat (PWL)

The paw withdrawal latency (PWL) to heat was explored using a Hargreaves box (IITC Life Science Inc., Woodland Hills, CA, USA) [28]. Here, rats were placed on a pre-warmed glass plate (30 °C), covered by transparent plastic boxes (dimensions 15 × 20 × 10 cm, H × W × H) (Figure 1B). After 15 min of habituation, a radiant heat source was applied to the plantar aspect of the right hind paw. The intensity of the halogen lamp was adjusted to 17%. The latency to hind paw withdrawal was measured with a cutoff time set to 20 s. Five trials with 5–10 min intervals were performed to calculate the mean PWL to heat stimuli.

### 2.5. Voluntary Pain-Related Behaviors

#### 2.5.1. Non-Evoked Pain Assessment (NEP)

Non-evoked pain was determined by comparing the weight-bearing (print area) of the affected (ipsilateral) and non-affected (contralateral) paw at rest (Figure 1B) [16]. For this purpose, we adapted the NEP for mice to rats.

Briefly, rats were separately placed in transparent boxes (dimensions 15 × 20 × 10 cm, H × W × H) on a 1-cm-thick and green light-illuminated glass plate. The boxes were covered by a slim LED panel (illuminated in red) to enhance contrast. Without prior habituation, images of the footprints of rats were captured at intervals of 30 s for a total period of 10 min. The areas of illuminated footprints of both hind paws were blindly determined on 10 different pictures for each rat using ImageJ [29]. The ratios of ipsilateral to contralateral illuminated hind paw areas were calculated for each time point and averaged for every animal. The image selection was based on predefined exclusion criteria, such as visible grooming, rearing, or an unsharp hind paw due to movement. The reduction in area ratios represents the degree of guarding behavior of the affected limb at rest.

#### 2.5.2. Movement-Evoked Pain Assessment (MEP)

Movement-evoked pain was assessed using the commercial CatWalk XT System (Noldus Information Technology, Wageningen, The Netherlands) [30,31] (Figure 1B). Only completed runs within the defined velocity range between 10 and 20 cm/s with a speed variance <60% were accepted as passed runs and included in the analysis. These inclusion criteria ensured comparability across all trials. The individual footprints were visualized by green light emitted into the glass plate on which the rats were running. Runs were recorded by a high-speed camera (100 fps) underneath the plate. Subsequently, three passed runs for each rat and time point were semiautomatically analyzed for two selected static (print area and stand duration) and dynamic (swing speed and stride length) gait parameters, which changed in different unilateral pain models by use of the CatWalk XT software:Print area: area of the whole pawStand duration (s): duration of ground contact for a single pawSwing duration (s): duration of any swing cycle of a single pawSwing speed (cm/s): rate at which a paw is not in contact with the glass plate

#### 2.5.3. Home-Cage Monitoring (HCM)

Pairs of rats were kept in custom-made cages for HCM over the whole observation period. HCM cages have been designed in cooperation with the technical workshop of the faculty of medicine of the WWU (Münster, Germany). They were designed according to the dimensions of the “SEALSAFE PLUS Rat-GR1800 DOPPELDECKER” (Tecniplast, Hohenpeißenberg, Germany) (2 levels, 3D enrichment, internal height of 38 cm, 1800 cm^2^ volume). As they were built of entirely transparent material, video recording via two cameras for night observations (the first two hours in the dark phase) was possible both from the top and in front of each cage (Figure 1B).

Videos were collected in an automated fashion and randomized to experimenters. Two blinded experimenters observed and rated videos using a specific ethogram and INTERACT software (Mangold International GmbH, Arnstorf, Germany) and finally statistically analyzed them. The specific ethogram depicts a wide range of individual (e.g., food/water intake, bipedal stance (BS)) and social (e.g., social resting (SR)) behaviors of rats kept under home-cage conditions.

### 2.6. microCT Visualization

The ipsi- and contralateral tibiae from each animal were scanned by micro-computer tomography (µCT) (Figure 1C) using a SkyScan 1176 (Bruker, Kontich, Belgium) after euthanization. Scans were performed at an isotropic resolution of 8.9 µm with a source voltage of 65 kV and a source current of 385 µA. Images were obtained at an angle shift of 0.5° with a 1.065 s exposure time using a 1 mm aluminum filter. To reduce artifacts, three pictures per angle were averaged. Pictures underwent axial reconstruction using the NRecon software (Bruker, Kontich, Belgium) for further evaluation. For scoring, shadow projections of each reconstruction were created using the CTVox software (Version number: 3.2.0 r1294, Bruker, Kontich, Belgium). Each tibia was orientated in the same manner, and the same transfer function for the opacity of the projection was used to visualize equal bone densities.

Tibial projections were scored by two independent experimenters in a blinded manner to determine bone destruction. An ordinal scale has been defined in advance to represent the status of bone destruction: no morphological changes compared to the non-treated control: 1; Slight lysis of the bone without loss of integrity: 2; Moderate bone lysis with loss of the overall shape but bone fragments still connected to the main body: 3; Severe fracture of the bone with visible free bone fragments: 4.

### 2.7. Data Analyses

The prior sample size calculation was based on the reporting effect size (2.74 standard deviations (SD), 95% confidence interval (CI)) from a systematic review [3] and current narrative [26] review with CIBP as a topic. PWT raw data was analyzed by nonparametric analysis, such as the Friedman test for within-group comparisons and the Kruskal–Wallis test for between-group comparisons. For PWL, NEP, MEP, and HCM behavior parameters, two-way ANOVA was used to analyze groups (to pre-value) and for between-group analysis. Multivariate behavioral data were analyzed by principal component analysis (PCA) with prior standardizations. PC selection was based on the largest eigenvalues. The first two principal components were plotted as biplots. Groups were added to the biplots for illustration but were not used during the PCA. The significance of group segregation was determined by multivariate analysis of PC loadings regarding the group with Tukey post hoc tests. Multivariate ANOVA (MANOVA) was performed to provide regression analysis and analysis of variance for multiple dependent variables by one or more factor variables or covariates.

A significance level of *p* < 0.05 indicates significant effects. Data were analyzed by Prism software, version 8 (GraphPad, San Diego, CA, USA) and SPSS (IBM, Armonk, NY, USA).

### 2.8. Study Design

#### 2.8.1. Cohort 1

Cohort 1 (*n* = 25 ♂, *n* = 25 ♀) contained three different experimental groups (Naive, sham, CIBP). For all animals of Cohort 1, behavioral assessments, including PWT, PWL, NEP, and MEP were performed before (pre) and on 3, 6, 8, 14, and 17 d after surgery. After euthanization, µCT visualization was performed for all animals (Figure 1D).

#### 2.8.2. Cohort 2: Home-Cage Monitoring

Cohort 2 (*n* = 32 ♂, *n* = 32 ♀) also contained animals from every experimental group and received behavioral testing before (pre) and 17 d after surgery. Additionally, the rats of Cohort 2 received home-cage monitoring, as they were kept in pairs in HCM cages for the entire observation period. Behavior was recorded during the first two hours of the night before (pre) and on day 3, 8, and 17 after surgery. Various combinations of the experimental groups (Naive–Naive, 4 Cages; sham–sham, 4 Cages; CIBP–CIBP, 4 Cages; shamBY–CIBP, 4 Cages) were used to assess social implementations of pain states. After euthanization, µCT visualization was performed for all animals of Cohort 2 as well (Figure 1D).

## 3. Results

### 3.1. Bone Cancer Causes Distinct Pain-Related Behavioral Trajectories in Rats of Both Sexes

Unilateral inoculation of Walker 256 mammary gland carcinoma cells into the right proximal tibia was used to induce bone cancer and investigate concomitant time-related behavioral changes assessed by multiple behavioral assays in Sprague Dawley^®^ (SD) rats of both sexes (Figure 1A). Our multimodal behavioral battery included traditional, reflex-based withdrawal assays on the hind paw, and approaches to assess voluntary pain-related and complex social and voluntary behavioral measures (Figure 1B). Reflex-based assays determined paw withdrawal thresholds (PWT) to mechanical stimulation with von Frey filaments and the paw withdrawal latencies (PWL) to heat stimuli. Voluntary pain-related behavior was assessed by a specific non-evoked pain (NEP), and movement-evoked pain (MEP) approach. Finally, rodent-specific complex behavior was determined by a home-cage monitoring (HCM) approach depicting a wide range of individual (e.g., food/water intake, bipedal stance (BS)), and social (e.g., social resting (SR)) behaviors of rats kept under home-cage conditions. Additionally, the body weight of the animals was determined to identify a possible influence of bone cancer progression on the animals’ fundamental survival functions. In the end, the cancer-induced bone destruction was evaluated using a micro-computed tomography imaging (µCT) approach (Figure 1C). To keep the interference between behavioral assays as small as possible, two separate cohorts were used: PWT, PWL, NEP, and MEP were assessed in cohort 1, while rodent-specific complex behavior in the home cage was assessed in cohort 2. Both cohorts included two control groups (naive, sham) and the cancer-induced bone pain (CIBP) group, which were examined for pain-related behavior before (pre) and at multiple time points (see timeline, Figure 1D) after cell innoculation.

Body weight in both sexes was unaffected by the devitalizing bone cancer in the CIBP group compared to both control groups (Figure 2A,B). In CIBP rats of both sexes, PWT was significantly decreased from 6 d until up to 17 d after cell inoculation compared to the pre-value. Exclusively in females, PWT at 6 d was reduced to the pre-value in the sham group (Figure 2A). PWL was unaffected in males but significantly decreased to the pre-value at 14 d and naive group at 14 d and 17 d in CIBP females (Figure 2A). A withdrawal response to lower mechanical force and shorter latency in withdrawal to heat stimuli on the hind paw surface indicates a state of hypersensitivity. NEP, represented by a reduction in the ipsilateral hind paw’s footprint (surface contact area) compared with the contralateral side at rest, was significantly reduced compared to the pre-value, and naïve and sham control groups from 6 d in males and 8 d in females until the end of the observation period (17 d) (Figure 2A,B). These reductions in the footprint area are typical signs of guarding injured limbs, also called pain avoidance behavior to mechanical weight-bearing [32]. MEP was assessed by gait analysis of static (print area, stand time) and dynamic (swing speed, swing time) parameters. A reduced print area, stand time, swing speed, and prolonged swing time, starting from 8 d in CIBP rats of both sexes, are characteristics of an antalgic gait pattern observed in unilateral rodent pain models (Figure 2A,B). However, significant print area reduction was observed in both sexes as early as 6 d. The progressive nature of bone tumor growth caused a manifestation of a bone cancer-induced pain phenotype via distinct pain modalities, including mechanical hypersensitivity, heat hypersensitivity (in females only), pain avoidance behavior, and an antalgic gait in the CIBP groups of both sexes.

Next, we asked which behavioral components of this pain phenotype are most relevant to group segregation over time and whether this behavioral signature changes as bone cancer develops. First, we performed correlation analyses of longitudinal trajectories for each pain-related behavioral outcome to determine which of the eight parameters assessed exhibited coherence concerning the underlying mechanisms (Appendix A). The correlation analysis was applied separately for sex and experimental groups, including the modality-specific time profiles. Significant correlations were determined for PWT and NEP with all MEP parameters in females, and PWT with NEP and the static parameters of MEP together with swing time in male CIBP animals (Appendix A). Interestingly, PWL only showed a positive correlation with NEP and stand time in CIBP females. In sham rats, a significant positive correlation was determined for PWT and print area in males and a negative correlation for swing speed and swing time in females.

Second, we performed two-dimensional principal-component analyses (PCA) to determine distinct factors that are primarily responsible for group segregation for each time point (Figure 2C and Appendix A, Appendix A) [33]. Principal-component (PC) scores and loadings revealed that no significant group segregation occurred in either sex before and 3 d post cell inoculation, as represented by unidirectional loadings and evenly distributed variance across both components (Figure 2C). In 6 d post-cell inoculation, CIBP males showed significant segregation from sham and naive groups, mainly along with the first principle component (PC1), as determined by a positive correlation between PWT, NEP, and print area (static gait parameter), associated with a negative direction for PWL. Progressive group segregation, driven by a positive correlation between PWT, NEP, and static and dynamic gait parameters (print area, stand time, swing speed), was established from 8 d for CIBP animals of both sexes. Except for PWL, swing time was negatively correlated to all parameters. The (negative and positive) correlations between gait parameters were representative of an antalgic gait pattern. PCA revealed a male phenotype characterized by mechanical hypersensitivity and an associated antalgic gait and guarding behavior beginning on 6 d. A similar phenotype, with additional PWL amounting to significant group segregation, could be identified in CIBP females from 8 d on.

### 3.2. Cancer-Induced Bone Destruction Is Associated with Pain-Related Behavior

Next, we wondered to which extent bone destruction, as a morphological outcome of bone cancer, is associated with the pain phenotype and whether sex-specific differences are present. Bone morphology was assessed 17 d post-cell inoculation by post-mortem µCT images with a double-independent and blinded scoring (Figure 3A). We chose post-mortem imaging since recurrent anesthesia for longitudinal µCT scans represents a potential confounding factor for behavioral outcomes because of the potential for alteration of neuronal activity [34,35]. As a degree of agreement among independent raters, the inter-rater reliability (ICC) score was κ = 0.972 for bone destruction (κ was calculated according to guidelines from Cicchetti and Sparrow [36], 0.0 = poor to 1.0 = excellent). Significant bone destruction and positive correlation with all parameters, except PWL and body weight, were identified in both sexes 17 d post inoculation (Figure 3B and Appendix A). PCA analysis revealed that the bone score significantly segregated controls from the CIBP group and is negatively correlated with pain-related behavior outcomes, except for swing time and body weight in both sexes (Figure 3C,D) and PWL in males (Figure 3D). Comparing both sexes revealed two main clusters in the direction of PC1, which were segregated by pain phenotype and bone destruction (Figure 3E). On the PC2 axis, these clusters were sex-specifically segregated by the presence of PWL in females and the higher body weight in males. The correlation between radiologically determined bone destruction, as a sign of progenerated bone cancer, and pain phenotype (e.g., PWT, PWL, NEP) (Figure 3E) underpinned a causality for the CIBP phenotype and bone destruction in both sexes at 17 d post-cell inoculation.

### 3.3. Bone Cancer Alters Rodent-Specific Complex Behavior in Rats of Both Sexes in a Home-Cage Setting

The rats showed an ipsilateral pain phenotype, characterized by mechanical hypersensitivity, hind paw guarding, and an antalgic gait pattern in both sexes, induced by a tibial bone tumor. This raises the question of to what extent CIBP alters complex rat-specific behaviors and whether other measurable behavioral parameters, e.g., social interaction, might be sensitive to pain states as well. Therefore, we assessed rat-specific behavior in the second cohort of animals in a home-cage environment.

Chronic pain states alter human activity profiles. Therefore, it is reasonable to examine the effects of CIBP on the specific complex behavior of rats under home-cage conditions without external influences during two hours at the beginning of the dark phase of the light cycle [10,19]. In our video-based home-cage monitoring setup, two independent investigators rated resting behavior (i.e., individual and social resting behavior), ambulatory behaviors (bipedal stance, jump to 1st floor), aspects of social (allogrooming), and pain-related behavior (e.g., ipsilateral grooming) in a blinded manner (Appendix A). Two rats of the same sex and various combinations of the three experimental groups were housed in a home cage. Our approach included housing equally treated animals (naive-naive, sham-sham, CIBP-CIBP) (Figure 1 D) but also combined sham-treated with CIBP animals to investigate the social transfer of CIBP (see below). As with the assessment of µCT images, the inter-rater reliability of the two independent and blinded raters and thus the ICC for the respective behavioral categories was determined. The ICC for the different parameters is in the excellent range (0.75–1.00) [36] (Appendix A).

Individual resting was not affected by the housing combinations in sham-sham and CIBP-CIBP animals of both sexes compared with the pre-value (Figure 4A). Male rats in the naïve-naïve group showed significantly decreased individual resting at 8 d and 17 d compared with the pre-value. The duration of social rest was higher for females in the control groups than for males but was significantly different from males in the sham-sham combination at 3 d only (Figure 4B). In the CIBP-CIBP group, social resting increased with time in both sexes, reaching the significance level of the pre-value at 8 d only in males. Total resting time in the observed two-hour interval varied around 50%, except for a significant increase in the CIBP-CIBP group in both sexes, compared with the pre-value (8 d in males, 8 d and 17 d in females) (Figure 4C).

The bipedal stance was significantly decreased in CIBP-CIBP at 8 d and 17 d for males and 17 d for females (Figure 4D). Jumping to the 1st floor was unaffected for the most part but significantly reduced in sham-sham at 3 d and CIBP-CIBP at 8 d (Figure 4E). Self-grooming activity was unaffected by either bone cancer or housing conditions (Figure 4F). A sex-specific significant difference in food intake was detected in the naive-naive group (pre, 3 d, 8 d) and the sham-sham group (pre) (Appendix A). Social playing behavior is one of the earliest forms of non-mother-directed social behavior observed in mammals and has been observed to contain behavioral patterns related to social, sexual, and aggressive behavior with a high reward value [37]. Playing behavior was unaffected over the whole observation period in the sham-sham group in both sexes (Figure 4G). In males, playing behavior was significantly decreased in the naive-naive group at 3 d, CIBP-CIBP group at 8 d and 17 d, and mixed housing group at 17 d, compared with the pre-value. A similar trend was observed in the corresponding female groups. Allogrooming (i.e., grooming of the partner rat) was only significantly reduced in the CIBP-CIBP group at 17 d compared with the pre-value. Grooming, or licking the affected side, is discussed as a surrogate behavior of spontaneous (non-evoked) pain in different rodent pain models [22,38,39], but was not observed under our experimental conditions (Figure 4H,I).

Again, the question arose as to which parameters from the home-cage observation drove possible group segregation and which were redundant. Therefore, PCA was performed for each time point to detect a linear correlation in both sexes (Appendix A). CIBP-CIBP males were characterized by increased individual and total resting time (and ipsi-grooming), reduced bipedal stance, jumping, and allogrooming. 

### 3.4. Social Transfer of Pain-Related Behavior from CIBP Rats to Sham Bystanders and Alterations in Complex Behavior Caused in Both Sexes

Recently, a phenomenon called “social transfer of pain” in male rodents has been characterized. This term describes the emergence of pain-related behaviors in sham-treated bystander rats (shamBY) following social interaction with a cage mate experiencing pain [40,41,42]. We hypothesize that such transfer of pain-related behavior also occurs when one rat bears a bone tumor (CIBP) and the other is shamBY. Furthermore, the transfer should also be detectable in females and reflected by changes in complex behavior in shamBY rats. Therefore, we investigated whether there was a transfer of behavior from CIBP rats of both sexes to shamBY-animals and which behaviors were affected. Mixed housing resulted in significant hypersensitivity to mechanical stimuli (Figure 5A), but not to heat (Figure 5B), in both paws of shamBY rats of either sex, compared to equal sham-sham housed animals (Appendix A). In contrast to shamBY rats, no significant mechanical hypersensitivity of the contra- or ipsilateral limb could be detected in sham or naïve rats. Equal housing of CIBP rats (CIBP-CIBP) caused ipsilateral but not contralateral mechanical hypersensitivity in both sexes. NEP was observed in CIBP rats of both sexes but not in shamBY rats at the ipsilateral site (Appendix A).

Individual resting was significantly increased compared to the pre-value at 17 d post-cell inoculation in CIBP males housed with a shamBY and vice versa (Figure 4A). Mixed female cage mates showed a similar, but non-significant trend in individual resting compared with the pre-value. Social resting was significantly reduced at 8 d and 17 d in males and for shamBY-CIBP combinations at 17 d in females, compared with the pre-value (Figure 4B). Total resting was decreased in the female shamBY-CIBP group to the pre-value and to that in males (Figure 4C). As with the CIBP-CIBP rats, the CIBP-shamBY group shows a reduction in bipedal standing of tumor-bearing rats in both sexes (Figure 4D). In contrast to the sham-sham group, jumping was significantly increased in shamBY at 3 d in males. Ipsilateral grooming of the tumor-bearing site was significantly increased at 8 d in CIBP-shamBY females. In the mixed housing groups, the question emerged as to which of the two animals initiated social vs. individual rest phases. Therefore, we investigated how often CIBP rats approached shamBY rats over time and vice versa (Appendix A). No significant changes were observed in this behavior for both sexes.

Although multidirectional loadings in PCA analysis were present, significant group segregation of tumor-bearing animals (CIBP) and shamBY from sham was detected in females at 8 d and 17 d, and in males at 3 d, 8 d, and 17 d (Appendix A, Appendix A). Strikingly, the shamBY rats significantly differed from the sham-sham husbandry in males from 3 d and significantly in females from 8 d (Appendix A, Appendix A).

PCA analysis of combined pain-related (which can be measured separately on both hind paws) and complex behaviors revealed that shamBY males are characterized by a mechanical hypersensitivity of both paws and reduced playing behavior along the PC1 axis, combined with increased individual resting (PC2) (Figure 5C). In contrast, the shamBY females were determined by increased playing behavior and decreased social and total resting (Figure 5D). The CIBP males, regardless of the housing combinations (CIBP-CIBP, CIBP-shamBY), were significantly distinguished from the controls by a higher ipsilateral bone score (PC1) and a reduction in playing behavior, bipedal stance, jumping to the 1st floor, and PWT of the ipsilateral paw. CIBP females differed significantly from controls by pronounced bone destruction, increased playing behavior, and individual resting. Just like in CIBP-CIBP males, mechanical hypersensitivity (PC1) and decreased ambulatory behavior along the PC2 axis were observed.

## 4. Discussion

In this study, we combined traditional reflex-based assays with rodent-specific complex behavior assessments to comprehensively phenotype a clinically relevant pain model for cancer-induced bone pain (CIBP) in rats of both sexes. CIBP in rats leads to a distinct CIBP-related phenotype, including (1) mechanical hypersensitivity of the ipsilateral hind paw, (2) non-evoked CIBP-related behavior (3), and antalgic gait pattern in both sexes. Heat hypersensitivity was associated only with female tumor-bearing rats. Progression of tumor growth causes CIBP phenotype clustering beginning at 6 d after cell innoculation in males and 8 d in females. Furthermore, intra-tibia tumor development generates bone destruction, correlating with the pain-related phenotype but not with the alterations in body weight of both sexes. Rodent-specific complex behavior analyses revealed both sexes’ social and ambulatory cancer-induced behavior alterations. In mixed housing conditions (shamBY-CIBP), the prevalence of social-resting behavior shifted towards individual resting. Furthermore, mechanical hypersensitivity of both hind paws of shamBY rats occurred in both sexes without a radiological diagnosis of bone cancer. This hypersensitivity is indicative of a social transfer of CIBP-related behavior from tumor-bearing rats to the shamBY.

### 4.1. Bone Cancer Causes a Sex-Specific Pain Phenotype Associated with Bone Destruction

Studies of CIBP using different rodent models are increasingly performed [3]. However, preclinical pain research, in general, has been criticized because many models and methods used for phenotyping appear to be artificial and only partially representative of the clinical situation, leading to a translational gap [3,7,26,43]. Assessing multimodal pain-related behaviors is a topic of ongoing debate in pain research, with the prevailing view that multiple modalities must be analyzed to address pain as a multidimensional phenomenon and bridge the translational gap of rodent pain research [7,11,44]. Furthermore, preclinical pain phenotypes in rodents are determined by a limited number of modalities using traditional stimulus-evoked assays with withdrawal responses of hind limbs based as endpoints, mainly in male animals [3]. In contrast, a multimodal approach to treat CIBP is crucial in the clinical setting [8,9].

To mimic the clinical situation for CIBP, we inoculated Walker 256 rat breast gland carcinoma cells intratibially in rats of both sexes. For the generation of this CIBP model, to create a distinct pain phenotype, we considered, firstly, to minimize the severity for the animals as far as possible and; secondly, to use a donor cell bank to reduce in vitro cell phenotypic changes during passage and avoid extensive cell culture amplification [27]; and, finally, to use a well-defined cell number for inoculation. These are essential parameters for the generation of the CIBP model [3,45,46]. This CIBP model is characterized by less interlaboratory variability, absence of metastases to other bones, CIBP-related behavior in rats of both sexes, and tumor progression independent of age, weight, and the estrous cycle. The typical observation time in this model ranges from 8-to-18 days post-cell innoculation to minimize the severity and address ethical trade-offs [26].

Multimodal pain-related behavioral approaches in both sexes using this CIBP model are scarce but urgently needed. In particular, the further development of novel behavioral methodologies using video assessments [10,19,32] to, for example, reduce experimenter bias [20] and acknowledge that pain states might lead to changes in complex rodent behaviors, is needed [19,47,48,49]. Recent findings regarding the social transfer of pain states in this and other pain entities [40,42] represent essential cornerstones for study planning and performance [10] in the future. Multimodal assessment and multivariate analysis of extensive complex behavioral data sets allow detailed insight into the underlying mechanisms and define pain entity-dependent phenotypes. Additionally, identifying redundant and, therefore, replaceable pain-related behavior assays, especially for evoked assays (direct interaction with the animal by the experimenter), is essential for the refinement of future study designs to avoid unnecessary animal stress and maximize clinical evidence [10]. A detailed and, especially, multidimensional pain phenotyping allows for the classification of the model in terms of its clinical relevance and, thus, its “value” for translational research. On the other hand, both desired effects through, e.g., pharmacological interventions, such as the reduction of non-evoked pain, and undesired ones (side effects) can be identified in the pain-related phenotype context.

We observed stable development of ipsilateral mechanical hypersensitivity, guarding behavior (non-evoked), and antalgic gait pattern, but no body weight reduction in rats of both sexes. These results are consistent with other studies, although there are contradictory results on body weight, depending on methodological aspects such as cell concentration, species, strain, sex, housing conditions, and observation duration [3,26,46,50]. Additionally, there are sex-dependent contradictory results for heat hypersensitivity [26,51,52]. We did not detect heat hypersensitivity in male rats; however, detection was evident during the late tumor progression phase in females. Reasons for this may be found at different experimental levels indicating that the detection of heat hypersensitivity does not represent a meaningful behavioral outcome in this CIBP model [26]. Tumor progression is also associated with sensory nerve sprouting [53], and the extent to which this is related to the development of heat hypersensitivity remains unclear. Sex differences also exist in the onset of pain phenotype expression in tumor-bearing rats. A CIBP-related phenotype can be observed in males from 6 d and females from 8 d, which may be caused by a sex-dependent release of lipoxins and endogenous lipoxygenase-derived eicosanoids [54]. Direct evidence to explain this difference in the development of CIBP-related phenotypes is not yet available, but it provides support to the hypothesis that pain phenotypes are sex-dependent [55,56,57].

Osteolysis caused by bone cancer is one of the essential macroscopic findings in animal studies of CIBP [3,26]. Here, we were able to identify a direct relationship between bone destruction and the CIBP-related phenotype, but not for body weight in both sexes. Despite the clear findings of radiological bone destruction and behavioral changes due to pain states, no effects on physical development were detected, calling the impact of CIBP on food intake into question; which rodent-specific complex behavior is modulated by CIBP in the first place and are cagemates and sex possible variables influencing the outcome?

### 4.2. Social Interaction with a Cage Mate Suffering from CIBP Alters Rodent-Specific Behavior

We developed a housing environment with additional structures and vertical spaces to assess the complex behaviors of rats within their familiar home cage. The home cage was built to serve basic physiological and behavioral rodent needs, including resting, grooming, exploring, or engaging in a range of social activities (play or allogrooming) [58]. Dependent on housing conditions, rat behavioral patterns were altered by bone tumor progression. Thus, we showed that social resting was increased, but ambulatory behaviors (e.g., bipedal stance) were reduced in both sexes. Furthermore, a significant decrease in social behaviors associated with physical activity, such as playing or allogrooming, was observed in males. In all behavioral categories, the trajectories of females are comparable to those of males, except for allogrooming. The reasons may be manifold: the tumor-related behavioral changes occur later in females, indicating different mechanisms for developing the tumor itself and, consequently, the CIBP-related phenotype. Sex-specific fundamental differences in neuro-immune pathways play a role in acute and chronic pain states [59,60,61] but have not been studied in depth for CIBP in both sexes [3,46]. Pain-related behaviors, such as excessive grooming or licking of the tumor-bearing leg, are only occasionally observed here. Reasons for this may be that there is no acute nociceptive pain [39] or acute skin injury, and extensive licking of the skin in this chronic CIBP model is not beneficial or necessary for the animal [62]. Furthermore, chronic pain conditions are very energy-consuming, so reducing or altering behavior, especially play, exploration, courtship, and mating, prevent re-injury and ensures that resources are initially reserved for defense, which seems reasonable from an evolutionary perspective [63].

Social interaction is characterized by the exchange of signals and adaptation of sensory and emotional states of the object. These conserved evolutionary behaviors have also been identified in rodents [23,40,47,64]. Rodent studies demonstrated that a rapidly adopting sensory-affective mechanism exists in an animal housed with a diseased cagemate of the experimental group, regardless of the valence of the information (pain, fear, or pain relief) [40,42,65,66]. Furthermore, this effect is independent of whether the pain is acute, nociceptive, or chronic. There is evidence that rodents show empathy-like behavior, which might deliver an evolutionary advantage since, among other effects, empathic behavior in social contexts seems to reduce pain perception [23,66]. For the first time, we demonstrated a social transfer of pain-related behavior (here, mechanical hypersensitivity) caused by CIBP in rats of both sexes. Presentation of CIBP-related behavior to an unaffected conspecific cagemate of the same sex, here a shamBY rat, leads to their adoption of sensory-affective states expressed by bilateral mechanical hypersensitivity in the absence of radiological signs of bone destruction. Speculatively, the ultimate reason for this hypervigilance state could be that an increased willingness to adopt fight-or-flight behavior is necessary to protect one’s own and group-relevant resources by activating the defense cascade [67]. How the transfer of sensory and affective states to conspecific unaffected cagemates occurs in CIBP can only be speculated. In other acute and chronic rodent pain models, the transfer was triggered by social signals such as ultrasound vocalization or pheromone release [47]. Therefore, it can be assumed that similar dissemination in CIBP might exist.

### 4.3. Limitations of the Study

In this study, we use an established animal model in rats to identify a sex-specific pain-related phenotype of bone cancer. The limitations here are the small group size of shamBY rats in home-cage analyses, which shows an altered phenotype. However, these results provide a first indication of social pain transfer in this bone cancer model, the mechanisms of which should be the subject of further investigation.

### 4.4. Implications for Study Planning and Severity Assessment in CIBP

The study findings can be discussed in light of two additional aspects: (1) the study design of animal experiments and (2) the question of how to assess the severity of experimental procedures. With respect to the former, our video observations showed that the composition of cagemate pairs could directly influence sensory-affective parameters. Additionally, social transmission of pain was observed, which needs to be addressed systematically in the study design to avoid any biases on the experimental and analysis level. More specifically, this could either mean separating pain groups from control groups or systematically including these mixed groups in the study design [10]. Besides this, using video approaches offers several additional advantages: (1) Video observations are carried out within the familiar home-cage environment of the experimental subjects, and, hence, the recordings can cover species-relevant times (e.g., night-time for rodents). (2) The direct influence of the experimenter and handling-related stress is reduced. (3) Video observations allow tracking the animals’ complex behavior without subjecting them to an external and, hence, unfamiliar test apparatus. (4) They increase the chance of detecting even subtle effects that might otherwise be overlooked [68]. (5) Lastly, such novel approaches enable a thorough understanding of species-typical behaviors that are not achieved by more traditional approaches.

With respect to the latter, the data presented here can also be used to discuss the question of how to assess the severity of experimental procedures imposed on animals objectively. Our results show that this chronic pain model could only detect behavioral changes with significant personal and technical effort. This not only demonstrates how difficult it is to assess and classify the severity of procedures according to the national and European guidelines [69,70], it also underlines the need for the development and validation of tools and methods to adequately, objectively, and reliably assess the animals’ welfare under varying experimental conditions [70]. This way, animal suffering can be minimized, and ethical and scientific considerations can be addressed.

## 5. Conclusions

In summary, we demonstrated the impact of CIBP evoked-, non-evoked behavior and complex rat-specific behaviors in animals of both sexes. Thus, we have provided the behavioral groundwork for mechanism-triggered (pharmacological) studies on CIBP and indicated which effect modifiers exist for animal pain studies in CIBP and in general, and how these can be considered in the future. Furthermore, we demonstrated social pain transfer in a bone cancer model for the first time in rats of both sexes, resulting in mechanical and, in females, also heat hypervigilance in non-tumor bearing shamBY.

## Figures and Tables

**Figure 1 cancers-15-01565-f001:**
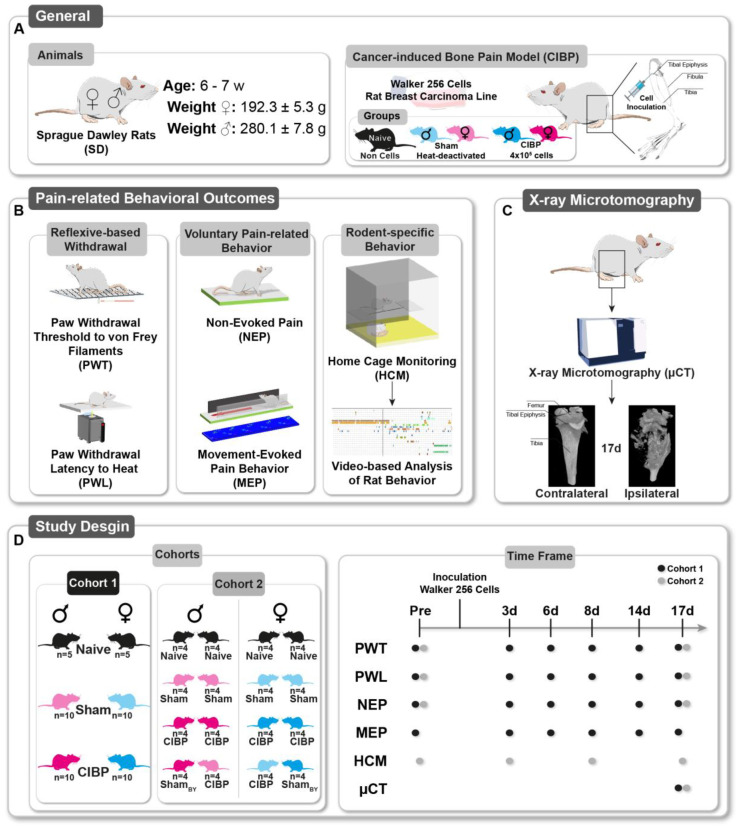
Study design. (**A**) Female and male Sprague Dawley rats (age 6–7 weeks, weight ♀ 192.3 ± 5.3 g (mean ± SD), ♂ 280.1 ± 7.8 g) were used. This study employed intratibial inoculation of Walker 256 cells as a surrogate for cancer-induced bone pain (CIBP). To assess the multimodal CIBP-related behaviors, rats were treated with Walker 256 cells (4 × 10^5^) or heat-deactivated cells (sham). Naive animals, without any manipulation (anesthesia or cell inoculation), were used to test the effect of intratibial cell injection. (**B**) Multidimensional pain-related behaviors were subgrouped into reflexive-based withdrawal, including punctate mechanical paw withdrawal threshold (PWT) to mechanical stimuli and paw withdrawal latency (PWL) to heat stimuli; voluntary pain-related behavior, including non-evoked pain (NEP) and movement-evoked pain (MEP) behavior; and rodent-specific behavior assessed by home-cage monitoring (HCM) of the first two night hours. (**C**) Ipsilateral and contralateral tibiae of all rats were examined for bone destruction by X-ray microtomography (µCT), post-mortem 17 days after cell inoculation. (**D**) Characterizing multidimensional pain behavior trajectories were assessed in a time-dependent manner with cohort 1 in both sexes. HCM was performed with cohort 2 in both sexes and different cagemate combinations.

**Figure 2 cancers-15-01565-f002:**
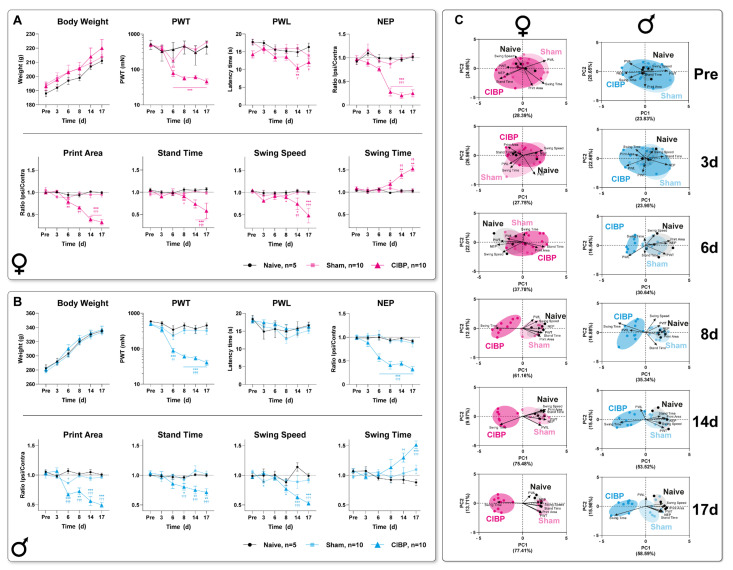
Bone cancer causes distinct pain-related behavior trajectories in both sexes of rats. (**A**,**B**) Trajectories of body weight and reflex-based withdrawal behavior, including paw withdrawal threshold to punctate mechanical stimuli (PWT), latency times to radiant heat stimuli (PWL), non-evoked pain-related behavior (NEP), and two static (print area, stand time) and two dynamic (swing speed, swing time) parameters of gait pattern analysis were determined in naïve, sham, and CIBP female (**A**) and male (**B**) rats. PWT was significantly decreased 6 d after cell inoculation in both sexes, and up to 17 d in CIBP rats. PWL was unaffected in CIBP males but significantly decreased in CIBP female rats at 14 d (to pre-value) and 17 d (to naïve). Assessment of the contralateral and ipsilateral hind-paw print area at rest revealed a non-evoked pain-related behavior (NEP) from 6 d in males and 8 d in females until 17 d, which was expressed by guarding behavior of the tumor-bearing hindlimb. Significant reduction to pre-value and naïve rats of print area, stand time, and swing speed combined with increased swing time, starting from 8 d in CIBP rats of both sexes, indicated an antalgic gait pattern. (**C**) Principal component analysis (PCA) of multimodal behavioral data was applied to identify a CIBP-induced phenotype in a time-dependent manner. Significant group segregation was determined at 6 d for CIBP males and 8 d for CIBP females. The results are expressed as mean ± SEM. Two-way ANOVA (repeated measures based on GLM) followed by Dunnett’s multiple comparison test. * for comparison to Pre; *p*-values: * ≤0.05, ** ≤0.01, *** ≤0.001. ^†^ for comparison to sham; *p*-values: ^†^ ≤0.05, ^††^ ≤0.01, ^†††^ ≤0.001. The PC components were selected to determine the eigenvalues. MANOVA was used for cluster analysis in PCA (see Appendix A). Black = Naïve rats, Light magenta = Female sham rats, Magenta = Female CIBP rats, Light cyan = Male cham rats, Cyan = Male CIBP rats.

**Figure 3 cancers-15-01565-f003:**
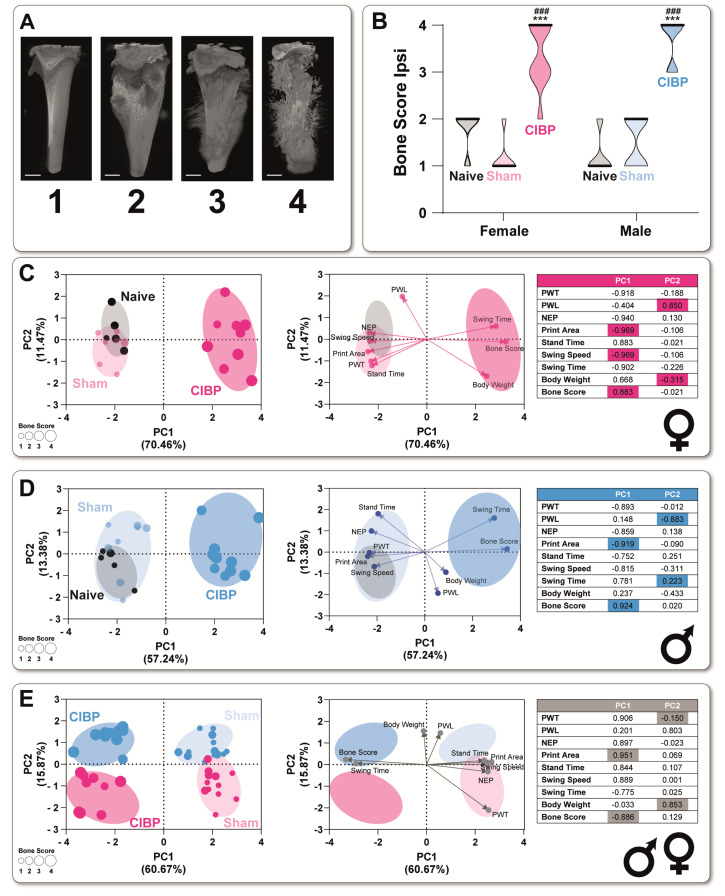
Cancer-induced bone destruction is associated with pain-related behavior. (**A**) Bone morphology was determined by post-mortem scoring of micro-computed tomography imaging (µCT) images from the ipsi- and contralateral tibiae. Scale bar = 5 mm (**B**) Scoring of the bone destruction of the ipsilateral side. In both female and male CIBP rats, a significantly increased bone score was detected, indicating increased bone destruction of the ipsilateral tibia. (**C**,**D**) PCA-assisted phenotyping showed significant group segregation of CIBP from control groups in both sexes, which was significantly triggered by the increased bone score and gait analysis parameters (see PCA loadings). (**E**) Including sex as a biological variable, the CIBP and sham control groups were segregated in both sexes based on PC1 axes (as in (**C**,**D**)). Sex segregation was determined by the significant PC2 loadings, body weight, and heat hypersensitivity (PWT), independent of the experimental group. Results in (**B**) are expressed as median ± 95%CI. Holm-Sidak’s multiple comparison tests followed two-way ANOVA. * for comparison to sham; *p*-values: *** ≤0.001. # for comparison to naive. *p*-values: ### ≤0.001. The PC components were selected to determine the eigenvalues. MANOVA was used for cluster analysis in PCA.

**Figure 4 cancers-15-01565-f004:**
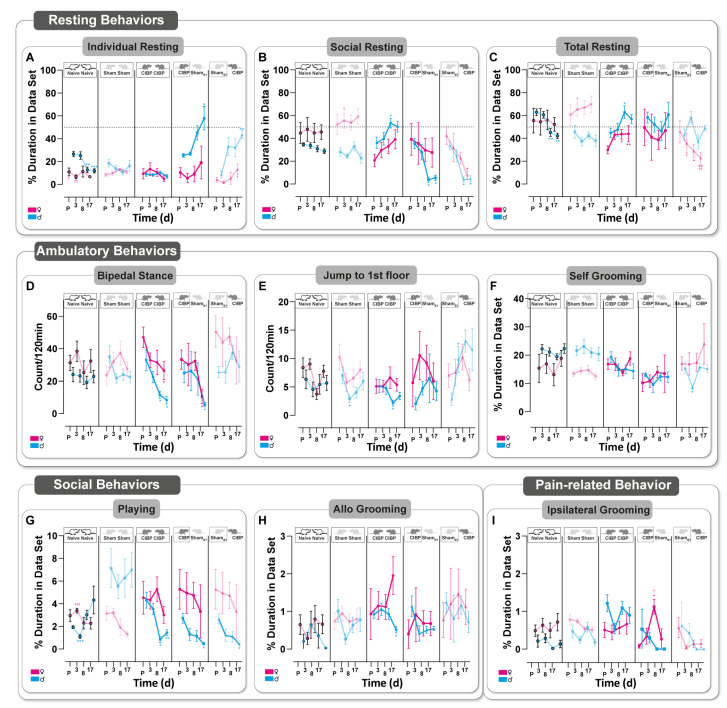
Bone cancer alters specific complex behavior in rats of both sexes. (**A**–**C**) Resting behavior was divided into individual (**A**), social (**B**), and total (**C**) resting. Significant differences to the pre-value are observed in the housing groups containing CIBP (CIBP-CIBP, shamBY-CIBP). (**D**–**F**) Ambulatory behaviors were divided into a bipedal stance (**D**), jumping to 1st floor (**E**), and self-grooming (**F**). The bipedal stance was significantly reduced in tumor-bearing rats (CIBP). (**G**,**H**) Social behaviors were analyzed by assessing playing behavior (**G**) and allogrooming (**H**). Playing behavior was significantly decreased in male housing combinations, including CIBP rats. Allogrooming was significantly reduced in the CIBP-CIBP male housing combination. (**I**) Grooming of the tumor-bearing (ipsilateral) hindleg is a proxy for pain-related behavior. A significant increase in ipsilateral grooming was determined on 8 d in CIBP females housed together with shamBY. Four different housing combinations are shown here. Each housing combination was repeated 4 times with other animals (*n*= 8, naïve; *n* = 8 sham; *n* = 8 CIBP; *n* = 4 CIBP-shamBY; *n* = 4 shamBY-CIBP). Boxes represent the data shown. Results are expressed as mean± SEM. Two-way ANOVA (repeated measures based on GLM) followed by Dunnett’s multiple comparison test * for comparison to sham; *p*-Values: * ≤0.05, ** ≤0.01, *** ≤0.001. ^†^ for comparison to male; *p*-values: ^†^ ≤0.05, ^††^ ≤0.01, ^†††^ ≤0.001.

**Figure 5 cancers-15-01565-f005:**
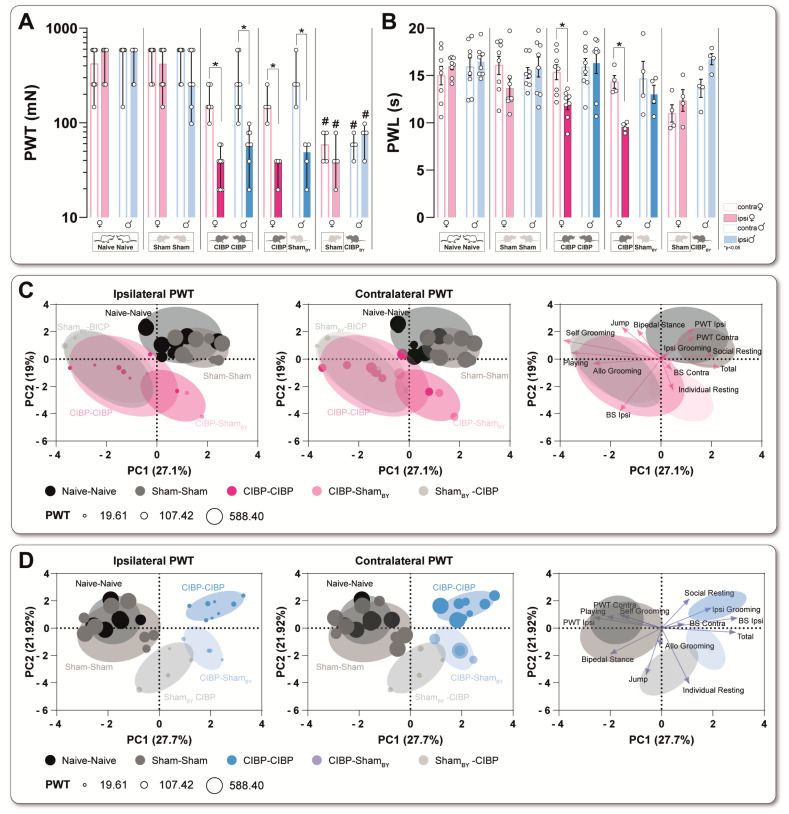
Social transfer of pain-related behavior to sham bystander (BY) rats caused by cancer-induced bone pain in both sexes (**A**) Mechanical (PWT) and heat (PWL) (**B**) thresholds of the ipsilateral versus contralateral hind paws in rats of both sexes in four different housing conditions 17 d after cell innoculation. (**C**,**D**) Representative visualization of the principal component analysis of female (**C**) and male (**D**) rats, including home-cage parameter, bone score, and mechanical threshold (PWT) of both hind paws at 17 d. The bubble size represents the PWT. Four different housing combinations are shown here. Each housing combination was repeated four times with other animals (*n* = 8, Naïve; *n* = 8 sham; *n* = 8 CIBP; *n* = 4 CIBP-shamBY; *n* = 4 shamBY-CIBP). Boxes represent the data shown. Results are expressed as mean± SEM. Two-way ANOVA (repeated measures based on GLM) followed by Dunnett’s multiple comparison test * for comparison to the contralateral site; *p*-Values: * ≤0.05, ^#^ for comparison to sham-sham; *p*-values: ^#^ ≤0.05.

## Data Availability

Data is contained within the article or Appendix A.

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
