# Peer review of "Behavioral Voluntary and Social Bioassays Enabling Identification of Complex and Sex-Dependent Pain-(-Related) Phenotypes in Rats with Bone Cancer"

_cancers, 2023, doi:10.3390/cancers15051565_

Round 1

Author Response

Thank you very much for evaluating our manuscript. 

Reviewer 2 Report

This is a well written manuscript on identifying behavioral outcomes in a rat model of cancer-induced bone pain. This is an important subject and the authors should be commended for their thorough study using both female and male rats and an extensive home cage monitoring system.

I have the following comments:

What was the humane endpoint, how was it monitored and did any of the animals reach this?

Figure 2: explain the colors/ lines

Figure 3: B what do the lines and individual figures represent

Figure 4: text for groups too small.

I do not understand the difference between CIBP-shamby and shamby-CIBP, this should be explained

In the results please consider the relevance of the information. E.g. it might be more relevant to say that there was no decrease in body weight in the CIBP groups instead of talking about the control groups.

What is the rationale of trying to correlate tumor destruction with body weight alterations? Which alterations?

Sometimes Walker-256 cells do not catch on or get suppressed. Could the results for the female rats be explained by some of the rats having almost no tumor burden – in other words did the uCT show that some rats had almost no tumor burden, and if so, maybe these rats should be excluded from the study? This might change the conclusions.

A major comment is the fact that a lot of statistical comparisons have been made on a limited number of animals. How do the authors account for multiple comparisons? There seem to be a lot of conclusions made on very low group sizes especially for the bystander effect (n=4)!

The evidence for the bystander effect is weak, it comes from a few animals in an unblinded experiment.

A section on the limitations of the study is missing.

Reviewer 3 Report

Congratulations to your manuscript. I see just minor mistypings.

Author Response

Thank you very much for evaluating our manuscript. After her valuable advice, a mother speaker has proofread the work. You will find changes in the manuscript. 
